# Future-Proofing Agriculture: De Novo Domestication for Sustainable and Resilient Crops

**DOI:** 10.3390/ijms25042374

**Published:** 2024-02-17

**Authors:** Ugo Rogo, Samuel Simoni, Marco Fambrini, Tommaso Giordani, Claudio Pugliesi, Flavia Mascagni

**Affiliations:** Department of Agriculture, Food and Environment (DAFE), University of Pisa, Via del Borghetto, 80-56124 Pisa, Italy; ugo.rogo@phd.unipi.it (U.R.); samuel.simoni@phd.unipi.it (S.S.); marco.fambrini@unipi.it (M.F.); tommaso.giordani@unipi.it (T.G.); flavia.mascagni@unipi.it (F.M.)

**Keywords:** new breeding techniques, genome editing, de novo domestication, next-generation sequencing, wild species, genes for tolerance to biotic/abiotic stresses, CRISPR/Cas9

## Abstract

The worldwide agricultural system confronts a significant challenge represented by the increasing demand for food in the face of a growing global population. This challenge is exacerbated by a reduction in cultivable land and the adverse effects of climate change on crop yield quantity and quality. Breeders actively embrace cutting-edge omics technologies to pursue resilient genotypes in response to these pressing issues. In this global context, new breeding techniques (NBTs) are emerging as the future of agriculture, offering a solution to introduce resilient crops that can ensure food security, particularly against challenging climate events. Indeed, the search for domestication genes as well as the genetic modification of these loci in wild species using genome editing tools are crucial steps in carrying out de novo domestication of wild plants without compromising their genetic background. Current knowledge allows us to take different paths from those taken by early Neolithic farmers, where crop domestication has opposed natural selection. In this process traits and alleles negatively correlated with high resource environment performance are probably eradicated through artificial selection, while others may have been lost randomly due to domestication and genetic bottlenecks. Thus, domestication led to highly productive plants with little genetic diversity, owing to the loss of valuable alleles that had evolved to tolerate biotic and abiotic stresses. Recent technological advances have increased the feasibility of de novo domestication of wild plants as a promising approach for crafting optimal crops while ensuring food security and using a more sustainable, low-input agriculture. Here, we explore what crucial domestication genes are, coupled with the advancement of technologies enabling the precise manipulation of target sequences, pointing out de novo domestication as a promising application for future crop development.

## 1. Introduction

Ten thousand years ago, human societies abandoned the nomadic lifestyle of hunters and gatherers and preferred permanent settlements and farming, leading to the birth of agriculture. Nowadays, the most widespread crops are the products of millennia of selection which occurred independently in several regions of the world: through selective breeding, wild plant species have been transformed into varieties with desirable traits, including architecture, flowering time, dormancy, fruit and seed size, yield, absence of antinutrients, thorns, and waxes, synchronous fruit ripening, and seed dissemination tailored to meet human needs. The selection of these traits is the so-called domestication syndrome in which plant survival depends entirely on human care [1]. Domestication has been considered a specific form of biological evolution [2]. It is a co-evolutionary interaction that leads to the creation of domesticated species for the benefit of another species. As a side-effect of selection, the development of reproductive barriers between some crops and their wild ancestors seemed to have favoured such a syndrome [3]. With the limited number of individuals of the progenitor plant species, much of the genetic diversity in the progenitor was left behind. Moreover, with each generation during the plant domestication process, only seeds from the best plants formed the next generation, causing a genetic bottleneck. This limited genetic diversity negatively affects crops by making them susceptible to many biotic and abiotic stresses [4,5].

Nowadays, global climate change poses a significant threat to current agricultural systems, raising concerns about crop resilience in extreme climatic conditions. Adverse environmental conditions for plant growth, such as nutrient deficiencies, poor soil conditions, drought, and pathogen attacks constitute the most relevant factors in agricultural yield reduction [6]. Developing more nutritious and resilient crops is necessary to mitigate the consequences of climate change, also from the perspective of an increasing world population in the coming years [7]. To identify highly resilient genotypes, breeders are actively screening crop and wild relative germplasms by exploiting cutting-edge omics technologies that have challenged traditional paradigms on genome evolution. Indeed, the advent of next-generation sequencing (NGS) technologies has facilitated the comparison of genome sequences between wild and cultivated plants, enabling the easier and faster identification of domestication genes (i.e., those sequences whose emergence in the evolutionary history of the crop has promoted the shift between wild and domesticated plants [8]). Indeed, recent progress in sequencing and bioinformatics has boosted the evolution of crop genomics from a single reference resequencing approach to embracing linear pan-genomes and, more recently, graph pan-genomes [9]. Many pan-genomes have been achieved for crucial crops, such as wheat [10], rice [11,12,13], tomato [14], and sunflower [15]. With the evolution of crop genomics, we have gained fresh perspectives on comparative genomics, and on the study of genetic diversity, domestication processes, breeding histories, and polyploidization events, standing at the precipice of a revolutionary era in assessing genetic resources.

In this global scenario, new breeding techniques (NBTs) are poised to become the agriculture of the future, introducing resilient crops to ensure food security despite of harsh climate events, all while safeguarding environmental sustainability. Within the goal of recovering missed variability from wild species, in the past ten years, the concept of de novo domestication has been developed [16,17]. In summary, the goal is to introduce favourable traits from the cultivated crop into wild- or semi-wild species. The introduction of favourable traits comprises four steps: (i) identification of genes involved in the domestication process and whose orthologs may be edited in a related wild or semi-wild species; (ii) performing genetic modification of these loci in the wild species through genome editing (GE) tools; (iii) screening for desirable genotypes and phenotypes; and (iv) agronomic evaluation of the new lines [18] (Figure 1).

According to Zhang et al. (2023) [19], de novo domestication can be divided into three main pathways depending on the starting genetic source: (1) re-domestication of the ancestors or wild relatives of crops; (2) de novo domestication of wild plants; and (3) accelerating the domestication of semi-domesticated plants or orphan crops.

Incorporating key traits from wild species into cultivated crops through conventional breeding approaches takes much work. Furthermore, transferring genes from distant relatives and exotic germplasm is extremely challenging due to many post-zygotic incompatibility hurdles. Hybrids produced through extensive hybridization combining distant and exotic germplasm are often sterile and require special techniques such as embryo rescue [20].

Precise mutation in domestication loci accelerates the domestication of wild species without compromising genetic background, yield, and resistance to biotic and abiotic stresses [18]. GE tools have proven to be an extremely useful approach. Clustered regularly interspaced short palindromic repeats/CRISPR-associated protein 9 (CRISPR/Cas9) is the most popular GE tool due to its high efficiency, versatility, simplicity, and cost. Briefly, CRISPR/Cas9 is a technology that enables the designed alteration of specific genomic sequences, producing DNA double-strand breaks (DSBs). Based on the endogenous DSB repair pathways, knock-out and knock-in can be performed [21]. A gene knock-out is usually caused by an error in template-free end-joining pathways, such as Non-Homologous End Joining (NHEJ), which results in a small deletion or insertion. Gene knock-in, on the other hand, requires either double-stranded DNA (dsDNA) or single-stranded DNA (ssDNA) donor templates, depending on the Homologous-Directed Repair (HDR) pathways, such as homologous recombination (HR), single-strand template repair (SSTR), and microhomology-mediated end-joining (MMEJ) [22]. However, a nuclease inactive Cas9 (deactivated Cas9 or nickases Cas9) linked with effector domains can be applied for activation/repression of genes [23], epigenome editing [24], base editing [25], and prime editing [26].

These NBT tools enable the precise introduction of mutations in target genes, including domestication genes, in a short period compared to traditional breeding methods, without altering the genome of the wild or semi-wild plants except at the target sites.

## 2. The Origin of Gene Domestication

Domestication has occurred repeatedly throughout the world, starting in many domestication centres [27,28,29] (Figure 2). For example, the wild ancestors of crops such as wheat, barley, and peas are traced back to the Fertile Crescent region [30,31,32,33,34,35,36]. Grains were cultivated in Syria as early as 9000 years ago [34], while figs were grown even earlier in the Jordan Valley about 11,300 years ago [37]. Although the transition from the wild gathering was gradual, the transition from a nomadic to a settled lifestyle is marked by the appearance of the first Neolithic villages with houses equipped with millstones for grain processing. The origins of rice and soybean cultivation date back to the same Neolithic period in China [12,38,39,40]. The world’s oldest known rice paddies, discovered in eastern China in 2007, revealed ancient cultivation techniques, such as flood and fire control [41]. Yam (*Dioscorea* sp.), pear millet, and sorghum were domesticated in Africa [42,43,44]. In Mexico, squash cultivation began about 10,000 years ago [45]. However, maize had to wait for selection by natural genetic mutations in its wild ancestor, a lowland wild grass known as teosinte (*Zea mays* ssp. *parviglumis*) [46,47]. While maize-like plants derived from teosinte appear to have been cultivated 9000 years ago, the first directly dated cob dates only to about 5500 years ago [1]. Maize then reached North America, where cultivated sunflowers began to be domesticated about 5000 years ago [48,49]. The beginning of potato cultivation in the Andean region of South America also dates to this period [50]. Jared Diamond (2002) [51] considers domestication the most significant technological development of the past 15,000 years (Figure 2).

By synthesising various perspectives, a comprehensive biological definition of domestication is a co-evolutionary process rooted in mutualism [53]. Domestication is a complex process that involves several key stages, including the beginning of domestication, the increased frequency of desirable alleles, the formation of cultivated populations, and eventually deliberate breeding, leading to changes between the progenitors and crops [1,54].

Genetic modifications have emerged through the iterative selection of plants exhibiting desired traits marking a clear distinction between domesticated taxa and their wild ancestors. Several large-effect significant loci mainly regulate the early morphological transformations observed during crop domestication. Genes controlling the critical morphological transformations in the early stages of domestication are considered domestication genes [55,56].

Genes associated with domestication, identified across diverse crops, exhibit conserved biological functions, positioning them as crucial candidates for de novo domestication endeavours through CRISPR/Cas-based GE. Examples of domestication genes include those found to affect seed shattering, which refers to the natural shedding of seeds when they become ripe, flowering, modifications of plant/inflorescence structures [57], the development of tuberous crops [50], and the dormancy of seeds in rice, tomato, and soybean [11]. Many key domestication genes of various species are enlisted in Table 1.

For instance, the *Q* gene plays a crucial role in the domestication of wheat. The genetic alteration from the *q* allele originated from a single amino acid mutation, which transforms the elongated speltoid spike into a subcompact form, making the glumes delicate and developing grains that can be easily threshed. In addition, *Q* pleiotropically affects a repertoire of other traits important for domestication, such as glume shape and tenacity, rachis fragility, spike length, plant height, and spike emergence time [58,59]. Similarly, in rice, *QTL of seed shattering in chromosome 1* (*qSH1*), *SHATTERING 4* (*sh4*) and *qSH3* loci contributed to the non-shattering of rice seeds in the early domestication [38,60,61]. In maize, *TEOSINTE BRANCHED 1* (*Tb1*) modification due to a retrotransposon insertion in the regulatory region contributed to change the plant architecture and photosynthesis through the inhibition of side branching and the phenotypic effect of alteration in the source–sink relations, leading to a yield enhancement [47,62]. *MULTIFLORA* (*S, an*), an allelic variation due to a missense mutation, is an example of a domestication gene that led to a modification of inflorescence development in tomato [63]. During the domestication of soybean, the loss of *Time of Flowering 12* (*Tof12)* function determined a decline in dormancy and seed dispersal, which may have predisposed the developing crop to latitudinal expansion, significantly impacting the adaptation of wild soybean during a phase of initial cultivation [64]. Recently, Goettel et al. (2022) [65] demonstrated that the CCT-domain gene *Protein, Oil, Weight, Regulator 1* (*POWR1*) mediates a large-effect protein/oil QTL in soybean domestication. Its CCT domain is truncated via the insertion of a transposable element (TE) that significantly contributes to raising seed weight, yield, and oil content while lowering protein content. Similarly, *SWEET1*, which encodes a bidirectional transporter pH independent of glucose, was suggested as an important domestication gene in grapevine (*Vitis vinifera*) [66]. Moreover, the retroelement, *Gret1*, has been identified as playing a key role in generating fruit colour variation in cultivated grapes due to its insertion into the promoter of *VvMybA1* [67]. In the evolution of *V. vifera sativa* from its wild ancestor *V. vinifera sylvestris*, a key role was the transition from dioecism to hermaphroditism that characterises the flowers of the cultivated grapevine. The results of Carrasco et al. (2020) [68] suggested that the *Ethylene overproducer-1* (*ETO1*) gene within the sexual locus region of the flower and the two genes located outside the sexual locus region encoding 1-aminocyclopropane-1-carboxylic acid synthase (ACS) and small auxin up RNAs (SAUR) proteins could be considered candidate genes for the control of sexual characters in grapevine.

**Table 1 ijms-25-02374-t001:** List of genes associated with domestication of some relevant crops.

Crop	Gene	Sequence Variation	Phenotype	References
Wheat	*APETALA 2-LIKE* (*WAP2, Q*)	Single amino acid variation	Free-threshing	[58]
	*REDUCED HEIGHT 1* (*Rht1*)	In-frame insertion	Semidwarfism	[69]
	*TENACIOUS GLUME* (*Tg1*)	-	Free-threshing	[70,71]
	*VERNALIZATION1* (*Vrn1*)	Mutation in the regulatory region	Vernalization response	[72]
Rice	*QTL of seed shattering in chromosome 1* (*qSH1*)	SNP in the 5′-regulatory region	Grain shattering	[60]
	*SHATTERING 4* (*sh4*)	Amino acid substitution	Grain shattering	[38,61]
	*SHATTERING ABORTION 1* (*shat1*)	Frameshift mutation	Grain shattering	[73]
	*PROSTRATE GROWTH 1* (*PROG1*)	SNPs in the coding region	Plant architecture	[74,75]
	*GRAIN INCOMPLETE FILLING 1* (*GIF1*)	Mutation in the regulatory region	Grain-filling	[76]
Maize	*TEOSINTE BRANCHED 1* (*Tb1*)	Retrotransposon insertion in the regulatory region	Plant architecture and photosynthesis	[47,62]
	*ZmSWEET4c*	SNPs in the promoter region	Grain-filling	[77]
	*TEOSINTE GLUME ARCHITECTURE 1*(*Tga1*)	SNP	Grain development	[78,79]
	*ramosa1* (*ra1*)	Mutation in the regulatory region	Plant architecture	[80]
	*LIGULELESS1* (*Lg1*)	Transposon insertion	Plant architecture	[81]
Tomato	*MULTIFLORA* (*S*)	Missense mutation	Inflorescence development	[63]
	*SUN*	Gene duplication mediated by retrotransposon	Fruit shape	[82]
	*fasciated* (*fas*)	Large insertion (6–8 kb) in the first intron	Locule number	[83]
	*locule number* (*lc*)	Two SNPs 1200 bp downstream a stop codon	Locule number	[84]
	*SELF-PRUNING* (*SP*)	Amino acid substitution	Growth habit	[85]
	*OVATE*	Premature stop codon	Fruit shape	[86]
	*FRUIT WEIGHT 2.2* (*Fw2.2*)	Mutations in coding and upstream region	Fruit size	[87]
Soybean	*Dt1* (ortholog of *GmTERMINAL FLOWER 1*)	SNP: substitution from Arg in the *Dt1* allele to Trp in the *dt1* allele at residue 166	Growth habit	[88]
	*Time of Flowering 11* (*Tof11*) and *Tof12*	-	Control flowering time at maturity	[64]
	*Protein, Oil, Weight, Regulator 1* (*POWR1*)	Transposon insertion	Grain-filling	[65]
Grapevine	*SWEET1*	-	Berry sugar content	[66]
	*Anthocyanin biosynthesis regulator in Vitis labrusca* (*VlmybA1*)	*Gret1* retrotransposon insertion	Fruit colour	[67]
	*Vvi AINTEGUMENTA-like* (*VviANT1*)	-	Fruit size	[89]
	*Ethylene overproducer-1* (*ETO1*)	-	Putative candidate gene for the control of sexual traits in grapevine	[68]

## 3. CRISPR as a Tool for De Novo Domestication

With the sudden population growth and climate changes, GE approaches can facilitate the development of new plant varieties that combine good performance with adaptability to stress-inducing environments and low-input management practices, contributing to a more sustainable farming system. Although plant domestication and genetic improvement have contributed to the present productivity level and quality traits in elite cultivars of crops, these processes have reduced genetic diversity and the loss of resilience to stress conditions in crops compared to their wild relatives [90]. Nowadays, breeders need to recover stress resistance key genes from wild species for their crops without changes to their genetic background. In this sense, de novo domestication using GE approaches can be useful.

Among the GE techniques, the “domestication” of the natural system CRISPR/Cas opened the windows for efficiency, versatility, simplicity, and a low-cost tool able to manipulate specific DNA/RNA sequences. CRISPR was discovered for the first time in *Escherichia coli* in the 1980s [91]. It is an adaptive immune system developed during the evolution by bacteria and archaea to defend themselves against virus attacks [92]. The defence components are mainly two: the endonuclease CRISPR-associated protein 9 (Cas9) and variants (e.g., Cas12 and Cas13), endonucleases responsible for cutting the foreign DNA/RNA, and a double-guide RNA (crRNA and tracrRNA) required to transport the Cas9 to a specific target DNA sequence. When a virus injects its DNA into the bacteria, the prokaryote CRISPR/Cas system cuts the virus nucleic acid in a DSB, rendering it inactive [21]. A review phrased CRISPR/Cas as the GE revolution [93] and could be a pioneering approach to recovering the plants’ “lost diversity” triggered by domestication.

Recently, researchers have highlighted that bacteriophages also possess the CRISPR system. The virus-encoded CRISPR/Cas system acts against other viruses during the same infection in a bacterium and even against the bacteria genome [94]. In this case, the endonuclease CasΦ is arranged in 700 to 800 amino acids residues, which is smaller compared to Cas9 (1000 to 1400 aa) [95,96], opening a new opportunity for a more efficient GE process in plant cells.

The GE revolution appeared when the double-guided Cas9 system had been simplified for application through the design of a chimeric single-guide RNA (sgRNA). In this way, the Cas9 protein can be delivered to any point of a genome and produce a DSB [97,98]. DSBs are the most deleterious type of DNA damage because they can result in the loss of large chromosomal regions [99]. Therefore, different cell repair complexes, such as NHEJ and HDR, act to reunite the ends of the double helix fragments. The NHEJ repair mechanism is prone to errors such as insertions, deletions, and substitutions. Differently, the accuracy of DNA repair is strongly enhanced by utilising the sequence from the sister chromatid or the homologous chromosome, constituting the basis of HDR [100].

Through DSB induced using CRISPR/Cas9, researchers around the world exploit NHEJ to induce short frameshift insertion–deletions in the sequence, disrupting the reading frame of mRNA by the formation of premature stop codons (knock-out) and HDR to introduce a specific mutation using a repair template (knock-in). Where reading-frame shift mutations are induced, the mutated transcripts derived from knock-out genes can be recognised and degraded by the nonsense-mediated mRNA decay (NMD) machinery or translated into truncated non-functional proteins leading to a loss of function mutation [101,102].

Over the past ten years, the CRISPR/Cas system has been continuously updated. Although the knock-out approach is the most widely used, catalytically impaired Cas variants (Cas9 nickases or deactivated, nCas9 and dCas9, respectively) fused with domains encoding effectors for activation/repression genes [23], epigenome editing [24], base editing [25], and prime editing [26], have been developed. In particular, regarding the de novo domestication approach, these two last techniques enable the recovery and copying of specific allelic sequences from elite crops into wild species, allowing specific nucleotide substitutions in target sequences. Base-editing techniques have significantly advanced in GE, enabling precise and effective single-base transitions or transversions at target sites without causing DSBs. There are three types of base editors: cytosine base editors (CBEs), adenine base editors (ABEs), and C-to g base editors (CGBEs) [103]. Cytosine base editors (CBEs) were among the first developed and effectively converted C:G to T:A base pairs. Initially, CBEs were engineered by fusing a cytidine deaminase rAPOBEC1 to the N-terminus of a dCas9 to improve efficiency [25] (Figure 3A).

This system deaminates the cytosine (C) to uracil (U), and the subsequent DNA repair and replication processes result in a C to T base conversion. Subsequent generations of CBEs, such as CBE2, CBE3, and CBE4 have been developed (Figure 3B–D). For these CBE versions, rAPOBEC1 [109], PmCDA1 [110], hAID [111], APOBEC3A [112], and evoFENRY [113] deaminases have been successfully applied in plants. In addition, to improve product purity and reduce involuntary indels, the CBE4 was implemented with a bacteriophage Mu Gam protein [104] (Figure 3E).

Adenine base editors (ABEs), with a similar structure and base-editing mechanisms of CBEs, are composed of nCas9 (D10A) linked with an adenosine deaminase. The complex converts adenine (A) to inosine (I), and after DNA repair and replication T:A to C:G base substitution is performed [107]. Updates of the first version have been made to increase the performance of the ABE system (Figure 3F–H).

The CBEs and ABEs systems induce only a base transition rather than base transversion. C-to g base editors (CGBE) use a rAPOBEC1 cytidine deaminase variant (R33A), nCas9 (D10A), and uracil *N*-glycosylase (UNG), an enzyme found in animals, plants and bacteria that can remove U from the DNA. Here, cytidine deamination transforms C to U, then UNG excises the U base created by the deaminase, forming an apurinic/apyrimidinic (AP) site that initiates the DNA repair process leading to C to G mutation (Figure 3I). Highly efficient C-to g editing has recently been reported in bacteria and mammalian cells [13,114]. Notably, the introduction of a N46L mutation in TadA-8e eliminated its adenine deaminase activity and resulted in a TadA-8e-derived C-to g BE (Td-CGBE) with highly efficient and precise C:G to G:C editing [108] (Figure 3J). However, the feasibility of Td-CGBE in plants has yet to be confirmed [103].

CBE and ABE for base transition have been optimised for editing efficiency and target regions, reducing off-target. On the other hand, for base transversion, such as CGBE for C to G and C to A, the editing efficiency is lower than other BEs in plants [115,116].

Moreover, one constraint that limits BEs’ application is the targeting scope of the BEs, which rely on the PAM requirements of Cas and the width of the catalytic reaction window [117]. To overcome this restriction, using different variations of Cas or engineered variants with altered or relaxed PAM specificities can be used to expand the editing scope. Although using some variants have extended the range of BEs, it greatly reduces its editing efficiency and increases its dependence on target sites. Thus, further research is needed to improve the efficiency of BEs in maintaining the recognition of relaxed PAM [103].

Prime editors (PEs), which can install desired base edits, small indels, and all kinds of single or multiple base(s) substitutions exploiting an RNA template without using donor DNA or DSB, have been used in plants and can accelerate crop improvement and breeding [118]. Prime editors are composed of nCas9 (H840A) linked with Moloney murine leukaemia virus reverse transcriptase (M-MLV-RT) at the C-terminus, and a prime editing guide RNA (pegRNA). The pegRNA is composed of three parts: a sgRNA targeting the specific site, a reverse transcription template (RTT) with an edited sequence, and a primer binding site (PBS) initiating reverse transcription [26]. Prime editing RNA guides nCas9 to target and the protein nicks the DNA. The reverse transcriptase reads the RTT and attaches the corresponding DNA nucleotides to the 3′ end of the nicked DNA. An endonuclease in the cell excides the fragment of the old single strand DNA target and seals the new reverse transcribed DNA nucleotides into the genome. Now, the target site has a mismatch with one edited strand and one unedited strand that is repaired, leading to an edited locus. The technique has evolved through several generations of improvements. The initial version, PE1, used nCas9 (H840A), and wild-type M-MLV-RT (Figure 4A).

To enhance editing effectiveness, PE2, PE3, PE4, PE5, and PEmax were developed (Figure 4B–F). In addition to the previous advancements, two more sophisticated techniques, the twin-prime editing methods twinPE and GRAND editor (genome editing by RTTs partially aligned [120] to each other but non-homologous to target sequences within dual pegRNAs) have been developed by using a pair of specially designed pegRNAs, in which the two RTTs were nonhomologous to the target sites but partially complementary to each other to enable the replacement of large indels or the insertion of specific genes (Figure 4G). TwinPE and GRAND editor are a DSB-independent form of GE that uses two prime editor proteins and two prime editing guide RNAs (pegRNAs) for programmable deletion, substitution, insertion, or inversion of larger DNA sequences within a genome [120].

Overall, prime editing outperforms conventional HDR strategies in terms of improving precise GE efficiency and overcoming the limitations of BEs techniques. It is a versatile and promising tool for precise GE because it allows for various base substitutions and the replacement of short indels in specific target genes of interest.

### 3.1. Trailblazing the Frontier: Pioneer Studies in De Novo Domestication of Plants through Genome Editing

The real concept of de novo domestication has been shown mainly in *Solanaceae* species and rice. Regarding *Solanaceae,* orphan crops, and wild tomatoes such as *Physalis pruinose* (‘ground cherry’) and *Solanum pimpinellifolium* (crop wild relative, CRW) have been used [121,122,123,124]. In particular, Li et al. (2018) [123] used a multiplex CRISPR/Cas9 strategy in *S. pimpinellifolium*, an ancestor of the modern tomato with excellent stress biotic and abiotic tolerance [125]. Coding sequences and *cis*-regulatory regions of *SELF-PRUNING* (*SP*), *SP5G*, *CLAVATA3* (*CLV3*), *WUSCHEL* (*WUS*), and *GDP-_L_-galactose phosphorylase 1* (*GGP1*) genes were edited to obtain compact plant architecture and synchronised fruit ripening, day-length insensitivity, enlarged fruit, and increased vitamin C level, respectively. These plants showed these desirable traits with improved disease resistance and salinity tolerance features [123].

Another approach in tomato cultivation, called the “two-in-one” strategy, has been used to accelerate breeding, combining elite cultivar male-sterility plants produced through CRISPR, with de novo domesticated plants. New varieties can be produced by backcrosses using F_2_ progenies as male and male-sterile elite as female. The first research with this approach was performed crossing a male-sterile elite line of tomato, Ailsa Craig (AC), edited for the *LESS ADHERENT POLLEN 3* (*SlLAP3*) gene, with the wild species de novo domesticated *S. pimpinellifolium* [126]. The wild plant was also edited for the *SP5G* and *SP* genes to create day-neutral germplasm with synchronised flowering and compact plant architecture [123]. After the cross, the F_1_ progenies exhibited typical intermediate phenotypes. In the F_2_ generation, plants with desirable traits including being stress tolerance from *S. pimpinellifolium*, and yield traits from AC were also selected. This strategy has proven to be an effective tool for accelerating the breeding process by leveraging desirable traits from a previously de novo domesticated plant, as well as incorporating traits such as high yield from elite cultivars acquired during conventional domestication.

Cultivated rice (*Oryza sativa*) is a diploid plant, and efforts applying de novo domestication have been made to exploit the benefits of wild rice allopolyploids, such as greater biomass, robustness, and vigour [127]. The authors developed an efficient transformation protocol [128], thus facilitating GE and high-quality genome assembly of *O. alta* (genome CCDD). Allotetraploid rice with the CCDD genome originated from a single hybridization event. In particular, the CC genome species (*O. officinalis* or *O. rhizomatis*) served as the maternal parent and an extinct species with the DD genome type served as the paternal donor [127]. The researchers collected seeds for 28 lines with the CCDD genome. *O. alta* (accession ID 2007-24 from National Nursery of Wild Rice Germplasm, Nanning, China) was chosen among other wild *Oryza* species and lines (eight *O. alta,* two *O. grandiglumis*, and eighteen *O. latifolia* lines) due to its callus induction and regeneration properties. The *QTL of seed shattering in chromosome 1* (*qSH1*), *Awn-1* (*An-1*), *SEMIDWARF 1* (*SD1*), and *GRAIN SIZE 3* (*GS3*) homologues genes have been edited through knock-out using the CRISPR/Cas9 system in both CC and DD genomes for shattering, awn length, shortened culm, and grain size, respectively. On the other hand, the *IDEAL PLANT ARCHITECTURE 1* (*IPA1*), *Grain number, plant height, and heading date 7* (*Ghd7*), and *Days to heading 7* (*DTH7*) homologues genes have been precisely edited using a BE technique for stem diameter, grain production, and day length to flower, respectively [127]. The results indicate that rapid de novo domestication and improvement of wild allotetraploid plants into stable food crops is theoretically feasible and technologically achievable in the near future.

Orphan crops (i.e., cassava, chickpea, cowpea, lentil and foxtail millet) may in the near future be the key to growing generally more resilient plants than major crops for the enhancement of marginal environments. Some of these orphan crops (such as cassava) are also beginning to be the subject of modern genetic improvement programs using genome editing techniques (issue reviewed in Ye and Fan (2021); Van Tassel et al. (2020); Yaqoob et al. (2023) [129,130,131]). In parallel, the attempt to de novo the redomestication of feral crops (i.e., feral cereal and feral Brassicaeae) by GE approaches also appears very interesting. Feral crops escaped cultivation with great potential in environmental adaptation [132]. Therefore, they contain valuable useful traits and retain some domestication characters. Due to their genetic closeness to crops, feral genotypes programs of neodomestication via genome editing techniques could be undertaken. Recently Pisias et al. (2022) [133] and Mabry et al. (2023) [134] reviewed these interesting issues.

### 3.2. New Delivery Techniques and Nanotechnology Advancements

Efficient regeneration and transformation plant protocols, which are major bottlenecks in the elite genotype of many species, are crucial for applying an efficient GE approach. Moreover, it was estimated that fewer than 0.1% of 370,000 higher plants in nature could be genetically manipulated [135]. Hence, these aspects must be considered before any GE project, especially using wild species. The most used method of plant genetic transformation is *Agrobacterium*-mediated delivery, which carries the foreign DNA into the plants using its transfer T-DNA, and particle bombardment using a gene gun, mostly in monocot species. Both methods cause random and stable integration of DNA into plant genomes. Another technique uses the protoplast transformation with plasmids expressing the CRISPR/Cas component or the direct insertion of ribonucleoprotein particles (RNPs) [136]. However, all these techniques are laborious with a long-term process. In addition, not all species and cultivars can be transformed and regenerated with this technique [137].

Recently, to overcome these challenges and obtain transgene-free Cas9 lines edited, other CRISPR/Cas9 delivery techniques, including viral vectors [138] and nanotechnology-based methods [139] have been developed to expand the possible wild plants to apply de novo domestication.

Virus-derived vectors have been reported as a potential alternative to express the CRISPR/Cas components, avoiding the tissue culture approaches required for stable transformation. These strategies are termed as virus-induced genome editing (VIGE) [140]. DNA viruses for GEs inevitably create the possibility of accidentally integrating foreign genetic material in the host genome, while RNA viruses develop their infectious cycles exclusively in the cytoplasm, thus resulting in plants free from foreign DNA, which should avoid raising regulatory and ethical issues [141]. To perform heritable mutations, the sgRNAs can be fused with RNA mobile elements, such as *Flowering locus T* (*FT*) to promote the mobility of reagents to apical meristems, inducing germline mutations [142]. The virus vector, including the sgRNA modified, is delivered through *Agrobacterium* infiltration into transgenic plants constitutively expressing Cas9 [143]. When utilising plants with a single copy of Cas9, this technique is especially interesting because transgene-free edited plants can be easily created through the segregation of the edited progeny. The results showed heritable bi-allelic mutations without evidence of virus transmission to progeny [144]. If no plants constitutively expressing Cas9 are available, nuclease must be delivered by virus. Depending on the genotypes and the viruses’ proliferation, CRISPR/Cas components can be accumulated to high levels within the host cell, leading to efficient and fast GE [141]. As vectors, RNA viruses have a limited cargo capacity, typically >1 kb, precluding their use for direct delivery of the Cas9 (more than 4 kb) [144]. Therefore, a dual RNA virus-based system could be performed, one carrying nuclease and the other the corresponding guide RNA [140].

As a different approach to simplify and speed up the GE protocol, there has recently been a raised interest in a graft-mobile gene editing system characterised by the production of transgene-free offspring in one generation, without the need for transgene elimination, culture recovery and selection, or the use of viral editing vectors [145].

Nanotechnology in agriculture is a rapidly emerging field, and nano carriers provide a promising approach for delivering biomolecules (DNA/RNA/proteins and RNPs) into plants [136]. Nanomaterials, with at least one dimension measuring less than 100 nm, can effectively penetrate the hydrophilic cell walls and lipid plasma membrane of plant cells, which have a size exclusion limit of 5–20 and 500 nm, respectively [146,147]. Single-walled carbon nano tubes (CNTs) and carbon dots can chemically functionalise to deliver genetic material into plant cell organelles, as shown in both nuclear and chloroplast genomes, without the need for external biolistic or chemicals and with no DNA integration into mature plants [148,149].

Looking ahead within the in-planta methods to obtain GE, foliar spray (or flower dip) with solutions containing CRISPR components can be another way to directly apply mobile CRISPR components [150,151].

Due to the development of these new techniques and materials, the delivery of GE reagents holds great promise for facilitating high-throughput plant genome engineering, including GE applied to de novo domestication.

## 4. Conclusions

The contemporary agricultural system faces the formidable challenge of feeding a steadily growing population, projected to reach 9.7 billion by 2050 (refers to the medium variant; United Nations, World Population Prospects: The 2019 Revision Population database, available at https://population.un.org/wpp/, accessed on 12 December 2023) [152]. Meeting this demand is complicated by diminishing cultivable land and the impact of climate change on yield quantity and quality. In the urgent quest for resilient genotypes, breeders turn to cutting-edge omics technologies, challenging traditional notions of genome evolution.

The domestication of crops, a transformative process that began millennia ago, has significantly shaped how we cultivate and rely on plants for sustenance. Unfortunately, early human agriculturalists likely sampled only a fraction of plants from natural populations, often leading to genetic bottlenecks associated with the selection of favourable phenotypes. These bottlenecks combined with a strong selection for high productivity traits, have undoubtedly reduced the effective population size of domesticated plants leading to increased genetic drift and decreased diversity within populations, particularly for the loss of genes involved in resistance to environmental stresses [153].

The consequences of this reduction in genetic diversity are now evident as global climate change poses unprecedented challenges to agriculture and the need for resilient crops with enhanced nutritional value has become more urgent than ever.

In response to these challenges, NBTs have emerged as powerful tools for revolutionising agriculture. Besides the possibility of easily identifying key domestication genes by using the last genomic resources, the CRISPR/Cas9 system has taken a central role among these tools. Its efficiency, versatility, simplicity, and cost-effectiveness make it a game-changer in genetic engineering. The ability to induce targeted mutations without altering the entire genome of the plant provides a precise and rapid way to introduce desired traits. CRISPR/Cas9 and its variants such as BE and PE offer an unprecedented chance for de novo domestication by introducing domestication genes in wild plants without altering their genetic background.

The fusion of traditional agricultural methods with advanced genetic technologies presents exciting prospects for the future of farming. The transition from ancient agricultural practices in the Fertile Crescent to the current GE techniques era marks a significant development in the ongoing relationship between humans and plants, holding promise for a more robust and sustainable food future.

## Figures and Tables

**Figure 1 ijms-25-02374-f001:**
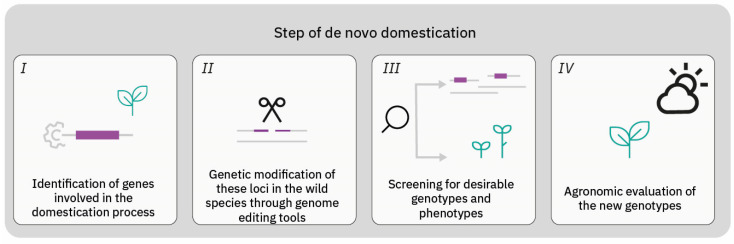
Strategy for de novo plant domestication. (I) The first step is the identification of genes involved in the domestication process supported by a high-quality reference genome sequence completely assembled and annotated. (II) The second step involves a highly efficient genetic transformation system to produce modification on orthologs gene in wild species through genome editing. (III) Edited plants are screened for desirable traits, such as resistance against biotic and abiotic stress. (IV) The last step is the agronomic evaluation of the new genotypes derived from wild species.

**Figure 2 ijms-25-02374-f002:**
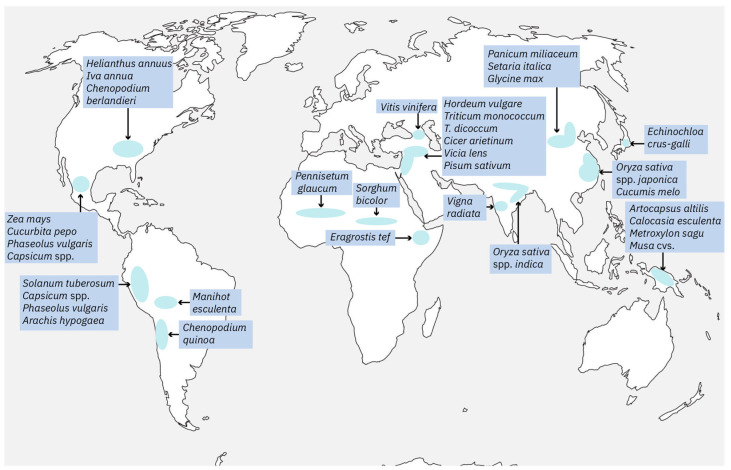
Map showing the major centres of crop domestication and some of the crops domesticated in each of them. Modified from Fuller et al. (2014) [52].

**Figure 3 ijms-25-02374-f003:**
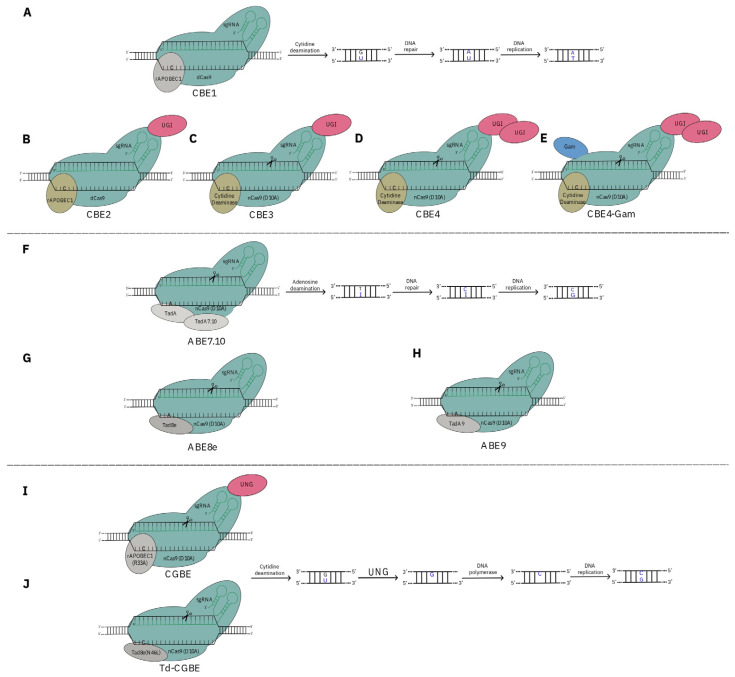
Graphical representation of clustered regularly interspaced short palindromic repeats (CRISPR)/nicking CRISPR-associated protein 9 (nCas9)-mediated base editing (BE), and its development. (**A**) The first generation of cytosine base editor (CBE1) is composed of the dCas9 (D10A/H840A) and a cytidine deaminase (rAPOBEC1). The cytidine deaminase catalyses the deamination of cytosine (C) in a narrow window and performs the base change from C to U (uracil) at a target site. After DNA repair, the guanine (G) is converted into adenine (A). U, recognised as thymine (T) during the DNA replication leading to a, is converted by U:A to T:A transition. (**B**) The second-generation cytosine base editor, CBE2, was engineered by fusing the CBE1 with a uracil DNA glycosylase inhibitor (UGI) to the C-terminus of dCas9 [25]. (**C**) The third-generation cytosine base editor, CBE3, was engineered by fusing different cytosine deaminases to the N-terminus of nCas9 (D10A), and fusing UGI to the C-terminus of nCas9 [104]. (**D**) The fourth-generation cytosine base editor, CBE4, is like CBE3 but with two UGI to the C-terminus of nCas9 [104]. (**E**) CBE4-Gam cytosine base editor was generated by adding bacteriophage Mu Gam protein to the N-terminus of nCas (D10A). (**F**) The adenine base editor (ABE7.10) complex is formed with nCas9 (D10A) and both wild-type adenine deaminase (TadA) and an evolved version (TadA7.10) of an adenosine deaminase. The adenosine deaminase catalyses an A to I (inosine) change at the target site, and during replication the original A is replaced with G. Finally, T:A to C:G conversion is performed in the non-target DNA strand [105]. (**G**) The ABE8e adenine base editor was used by fusing a more efficient protein TadA8e to the N-terminus of nCas9 (D10A) [105]. (**H**) The ABE9 adenine base editor was engineered by fusing TadA9 (TadA8e with V82S and Q154R mutations) to the N-terminus of nCas9 (D10A) [106]. (**I**) The C-to g base editor, CGBE, consists of cytosine deaminase, nCas9 (D10A) and uracil *N*-glycosylase (UNG). Cytosine deaminase modifies C to U, then UNG removes U base leaving an apurinic/apyrimidinic (AP) site that is repaired through an error-prone DNA polymerase leading to nucleotide changes including C to G substitution. The C:G to-G:C transversion occurs during DNA replication. As nCas9 (D10A) nicks the target strand, a DSB is formed when the abasic site on the non-target strand is converted into a nick by an apurinic or apyrimidinic site lyase (AP lyase). This DSB results in indel formation [107]. (**J**) The Td-CGBE base editor consisting of a TadA8e (N46L) and nCas9 (D10A) is the last highly efficient and precise C:G to G:C editing [108].

**Figure 4 ijms-25-02374-f004:**
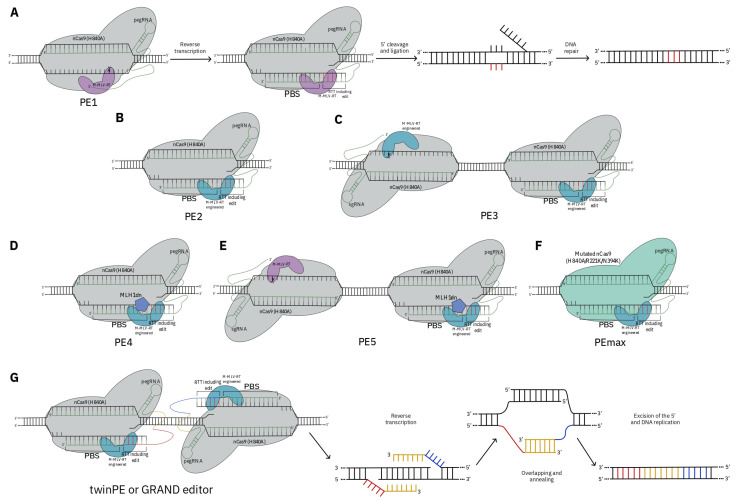
Graphical representation of clustered regularly interspaced short palindromic repeats (CRISPR)/nicking CRISPR-associated protein 9 (nCas9)-mediated prime editing (PE), and its development. (**A**) The PE system mainly consists of a nCas9 (H840A), a M-MLV-RT (Moloney murine leukaemia virus reverse transcriptase) linked to the N-terminus of the protein, and a prime editing guide RNA (pegRNA). The pegRNA is formed of three parts, including a single-guide RNA (sgRNA), which targets the specific site, a reverse transcription template (RTT) implementing the edit, and a primer binding site (PBS) initiating RT. The complex, nCas9 (H840A)-M-MLV-RT and pegRNA bind the target sequence, and the M-MLV-RT helps the 3′ DNA end from the PBS to prime the reverse transcription of an edit-encoding extension from pegRNA in the target site [26]. (**B**) The second-generation prime editor, PE2, was engineered by fusing a modified M-MLV-RT (H9Y/D200N/T306K/W313F/T330P/L603W mutations) to the N-terminus of nCas9 (H840A) [26]. (**C**) The third-generation prime editor, PE3, used another sgRNA-nCas9 to nick also on the opposite strand [26]. (**D**) The fourth-generation prime editor, PE4, added a dominant negative mismatch repair (MMR, sush us MSH2, MSH6, MLH1, and PMS2) protein (MLH1dn) [119]. (**E**) The fifth-generation prime editor, PE5, is an update of PE4 adding another sgRNA-nCas9 on the opposite strand [119]. (**F**) PEmax was created using a Cas9 that harbours three mutations (H840A/R221K/N394K) [120]. (**G**) TwinPE or GRAND editor used two pegRNA-nCas9-RT complexes. The two RTTs (red and blue line) partially aligned to each other (orange line). After reverse transcriptase activity, the single-strand DNAs (red and blue lines), bind to each other via their complementary ends. Following DNA replication and repair, the original 5′ flaps were replaced by annealed 3’ flaps containing the edited DNA [120].

## Data Availability

Data are contained within the article.

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
