# Peer review of "Future-Proofing Agriculture: De Novo Domestication for Sustainable and Resilient Crops"

_ijms, 2024, doi:10.3390/ijms25042374_

Round 1

Reviewer 1 Report

Comments and Suggestions for Authors

Rogo and co-authors present a well-reasoned argument for increasing the resiliency of crops by domesticating wild relatives and orphan species through the manipulation of key genes. Their description of the “domestication genes” selected during crop development is informative. By altering orthologs of these genes in wild species, the authors suggest that breeders could accelerate the production of crops with agricultural traits as well as stress resistance derived from new genetic backgrounds. Initial results that they describe on the application of this strategy in tomato and rice support their claim.

Gene editing is the technology the authors envision being used to alter specific genes in wild species to obtain agricultural traits. While their description of CRISPR and its evolving applications is excellent, the depth of this information seems somewhat out of place in this review. The major points of the review could be made without the level of technical detail that is included on gene editing.

 It is gratifying to see challenges to the implementation of this strategy discussed. The introduction of CRISPR components into plant cells and the regeneration of modified plants are not trivial. To date, the use of nanoparticles and genes that promote plant regeneration is promising, but these approaches have been applied primarily to model species. It should also be mentioned that, for the first time in human history, there is social pressure against the adoption of technology that could benefit agriculture.

Comments on the Quality of English Language

Author Response

Dear Reviewer,

The present submission is a revised version of the manuscript “Future-Proofing Agriculture: De Novo Domestication for Sustainable and Resilient Crops”, ID: ijms-2851693.

We thank you for the comment. We appreciate the suggestions and amend the text according to the suggestions. In the revised manuscript, the changes are highlighted in yellow.

Best Regard,

Ugo

Comments and Suggestions for Authors

Rogo and co-authors present a well-reasoned argument for increasing the resiliency of crops by domesticating wild relatives and orphan species through the manipulation of key genes. Their description of the “domestication genes” selected during crop development is informative. By altering orthologs of these genes in wild species, the authors suggest that breeders could accelerate the production of crops with agricultural traits as well as stress resistance derived from new genetic backgrounds. Initial results that they describe on the application of this strategy in tomato and rice support their claim.

Gene editing is the technology the authors envision being used to alter specific genes in wild species to obtain agricultural traits. While their description of CRISPR and its evolving applications is excellent, the depth of this information seems somewhat out of place in this review. The major points of the review could be made without the level of technical detail that is included on gene editing.

 It is gratifying to see challenges to the implementation of this strategy discussed. The introduction of CRISPR components into plant cells and the regeneration of modified plants are not trivial. To date, the use of nanoparticles and genes that promote plant regeneration is promising, but these approaches have been applied primarily to model species. It should also be mentioned that, for the first time in human history, there is social pressure against the adoption of technology that could benefit agriculture.

Reviewer 2 Report

Comments and Suggestions for Authors

Author Response

Dear Reviewer,

The present submission is a revised version of the manuscript “Future-Proofing Agriculture: De Novo Domestication for Sustainable and Resilient Crops”, ID: ijms-2851693.

We thank you for the comment. We appreciate the suggestions and amend the text according to the suggestions. In the revised manuscript, the changes are highlighted in yellow.

Best Regard,

Ugo

Comments and Suggestions for Authors

Abstract:

Very good and clear regarding the title.

Referred to all topics of the work.

Explains very well the importance of evolution and the great need for progress in plant breeding

The keywords are very well chosen.

  1. Introduction and 2. The Origin of Gene Domestication:

Very good, clear and coherent writing.

He explains all the research very well, based on his own experimentation and knowledge. To

confirm and make the document stronger, it cites 155 authors.

  1. Conclusions:

Very objective and clear.

Explains again the problem of plant domestication and the great challenge of feeding a steadily

growing population

It concludes by presenting the great utility the new breeding techniques (NBTs) are emerging as

the future of agriculture, offering a solution to introduce resilient crops that can ensure food

security, particularly against challenging climate events.

References:

The work cites 155 references and they are very diverse (years and authors).

The work is very strong and rich, as can be seen from the large number of references.

GENERAL INFORMATION:

Pay attention to the spacing between lines of figures 1, 2, 3 and 4.

Table 1 could also be improved with less spacing between lines and perhaps placing lines to make

it easier to read.

It is a very dense work and difficult to read, but it clarifies the reader very well as it has excellent

sequencing of information.

It is based on and very well supported by an excellent bibliography (which is very well placed in the

text).

NBT's and the CRISPR/Cas9 system are very important methods for the future of agriculture as

presented in this work, which is why I use your words "NBTs have emerged as powerful tools to

revolutionize agriculture. In addition to the possibility of easily identifying the main genes of

domestication using the latest genomic resources, the CRISPR/Cas9 system has assumed a central

role among these tools."

The article is very well written, very clear and is very useful for the international community,

so congratulations to the authors